# Local Invariance of Divergence-Based Quantum Information Measures

**DOI:** 10.3390/e27101051

**Published:** 2025-10-10

**Authors:** Christopher Popp, Tobias C. Sutter, Beatrix C. Hiesmayr

**Affiliations:** Faculty of Physics, University of Vienna, Währingerstraße 17, 1090 Vienna, Austria; tobias.christoph.sutter@univie.ac.at (T.C.S.); beatrix.hiesmayr@univie.ac.at (B.C.H.)

**Keywords:** quantum information, entropy, mutual information, divergence, local invariance, reversal channel

## Abstract

Quantum information quantities, such as mutual information and entropies, are essential for characterizing quantum systems and protocols in quantum information science. In this contribution, we identify types of information measures based on generalized divergences and prove their invariance under local isometric or unitary transformations. Leveraging the reversal channel for local isometries together with the data-processing inequality, we establish invariance for information quantities used in both asymptotic and one-shot regimes without relying on the specific functional form of the underlying divergence. These invariances can be applied to improve the computation of such information quantities or optimize protocols and their output states, whose performance is determined by some invariant measure. Our results improve the capability to characterize and compute many operationally relevant information measures with application across the field of quantum information processing.

## 1. Introduction

A central goal of quantum information theory is the precise quantification of correlations, uncertainties, and distinguishability within quantum systems. Various information-theoretic quantities have been defined to capture the fundamental limits of quantum information processing tasks like communication, computation, or entanglement manipulation.

Traditionally, the fundamental quantity is the von Neumann entropy [1], from which many quantities relevant in classical information science, like the mutual information or the conditional entropy, can be generalized to the quantum regime (cf. [2]). These measures are relevant due to their operational meaning in quantifying optimal rates of achieving various information processing tasks in the asymptotic regime of infinitely many independent uses of a resource (e.g., quantum state). A notable example is the quantum relative entropy [3], which can be related to quantum hypothesis testing [4]. Several other information quantities can be expressed in terms of the relative entropy, such as the private information for the rate of secret-key distillation and the coherent information for the rate of entanglement distillation [5]. In the so-called one-shot setting (cf. Ref. [6]), only a finite amount of resources are considered. In this approach, the goal is to find optimal rates to transform the limited resources, such as quantum states, into the target states while allowing fixed error bounds. Moreover, the one-shot setting is crucial for security analysis in the context of quantum cryptography without any assumptions on the actions of a malicious third party [7]. As in the asymptotic setting, several information quantities with operational meaning have been identified such as the ε-hypothesis testing mutual information [8] and the smooth max-mutual information [9] that can be related to secret-key distillation in the one-shot setting.

As quantum technologies advance, the landscape of information quantities and related processing tasks becomes increasingly diverse (see Ref. [10] for a comprehensive overview of information measures and related processing tasks) and a unified framework capable of accommodating both asymptotic and one-shot scenarios is essential. Generalized divergences offer such a unifying language. By definition, each divergence satisfies monotonicity under completely positive trace-preserving (CPTP) maps, i.e., a data-processing inequality under the action of quantum channels, that guarantee operational meaning and allow us to derive an entire spectrum of information measures. Examples include the Petz–Rényi relative entropy [11,12,13], the sandwiched Rényi relative entropy [14,15], the geometric Rényi relative entropy [16,17] and (smooth) max/min-relative entropies [7]. Note that the relative entropy also satisfies a data-processing inequality and thus many information quantities in the asymptotic setting can be expressed by this specific divergence. Replacing the relative entropy by other divergences in similar expressions allows one to define generalized quantities that are applicable in other regimes, e.g., the one-shot setting.

Despite their broad applicability, evaluating these divergence-based measures poses significant computational problems in the case of high-dimensional quantum states or if their definition involves challenging or infeasible optimizations. In this work, we define several types of divergence-based information measures and prove their invariance under local isometric or unitary transformations.

In Section 2, we introduce the setting and notation and define the types of generalized divergence-based information quantities. We continue to present and prove the main result of this contribution, namely the invariance of information quantities of the defined types under local isometric or unitary transformations in Section 3. Finally, we conclude our results in Section 4 and provide an outlook to potential applications and future research directions.

## 2. Methods: Generalized Divergences and Information Quantities

In this section, we briefly introduce the notation and necessary objects to define several types of generalized information quantities and prove their invariance under local isometric or unitary transformations.

### 2.1. Notation and Setting

We consider two parties *A* and *B* with corresponding Hilbert spaces H=HA⊗HB. L+(H) and D(H) denote the spaces of positive-semidefinite operators and density operators (i.e., quantum states) acting on H, respectively. Isometries are denoted by a capital *V*. Corresponding channels, i.e., completely positive and trace-preserving maps, are written as V(·)≡V(·)V†. Unitary operators are referred to with a capital *U*. The identity map is written as id and ∘ denotes the composition of maps. States or operators are labeled with the systems they act on. For any multipartite state or operator, e.g., the state ρAB, the marginal state or operator is given by ρA:=TrB[ρAB].

### 2.2. Generalized Divergence-Based Types of Information Quantities

The information quantities we analyze in this contribution are defined via so-called generalized divergences D [18,19].

**Definition** **1**(Generalized divergence, data-processing inequality)**.**
*For quantum states ρ∈D(H) and positive-semidefinite operators ζ∈L+(H), a function D:D(H)×L+(H)→R∪∞ that satisfies the data-processing inequality under any channel N, D(ρ||ζ)≥D(N(ρ)||N(ζ)), is called (generalized) divergence.*

For quantum states, the data-processing inequality can be interpreted as the property that no physical transformation can make two states more distinguishable. It is known that any divergence is invariant under the application of isometric transformations V [10]:(1)D(V(ρ)||V(ζ))=D(ρ||ζ). Generalized divergences are used to define generalized information quantities for bipartite quantum states ρAB. There exist several ways to define information quantities based on divergences with operational meaning. In this work, we consider quantities that relate a bipartite state and its local marginals. Using specific instances for the divergences, these quantities can be related to various information-processing tasks (cf. [10] for a comprehensive overview). The method to prove local invariance of the quantities we present in this work, however, does not depend on the specific form of the divergence, but only on how it is applied to the state. We therefore define several types of information quantities independent of the specific form of the divergence. Some types involve a so-called smoothing, i.e., an optimization over an environment of the quantum state. For this, the sine distance [20] is often used as a distance measure that is closely related to the fidelity [21] as defined below.

In the following, let ε∈[0,1] and let ρ and σ be states.

**Definition** **2**(Fidelity, sine distance)**.**
*The fidelity of two quantum states ρ and σ is defined as follows:*(2)F(ρ,σ):=Tr[σρσ]2
*The sine distance is defined as:*
(3)P(ρ,σ):=1−F(ρ,σ)

However, other distinguishability functions potentially with different properties can be used as well. In this work, we only require that the function obeys the data-processing inequality and invariance under isometric transformations. As both properties are satisfied by any divergence (cf. Definition 1 and (Equation 1)), including the sine distance [10], the smoothing environment Bε(ρ) around a state ρ is defined as follows:

**Definition** **3**(Smoothing environment)**.** *Let D be any divergence.*(4)Bε(ρ):=ρ^:D(ρ||ρ^)≤ε

For the sine distance, it specifically reads:(5)BPε(ρ):=ρ^:P(ρ,ρ^)≤ε=ρ^:F(ρ,ρ^)≥1−ε2. In this work, we define and analyze the following types of information quantities:

**Definition** **4**(Types of generalized mutual information)**.**
(6)I1(ρAB):=D(ρAB||ρA⊗ρB)
(7)I2(ρAB):=infσBD(ρAB||ρA⊗σB)
(8)I3ε(ρAB):=infρ^AB∈Bε(ρAB)σBD(ρ^AB||ρA⊗σB)
(9)I4ε(ρAB):=infρ^AB∈Bε(ρAB)D(ρ^AB||ρA⊗ρ^B)

**Definition** **5**(Types of generalized conditional entropies)**.**
(10)H1(ρAB):=−D(ρAB||1A⊗ρB)
(11)H2(ρAB):=−infσBD(ρAB||1A⊗σB)
(12)H3ε(ρAB):=−infρ^AB∈Bε(ρAB)σBD(ρ^AB||1A⊗σB)

The defined types include information quantities in both the asymptotic and the one-shot setting. Examples of this are the (generalized) quantum mutual information (I1,I2), the coherent information (H2), the Petz-Rényi mutual information, the sandwiched Rényi mutual information, the geometric Rényi mutual information, the smooth min-mutual information (see Ref. [22] for these quantities), the ε-hypothesis testing mutual information (I3ε), the smooth max-conditional and min-conditional entropy (H3ε), and the smooth max-mutual information (I4ε).

## 3. Results: Local Invariance of Information Quantities

We show that the types I1,I2 and I3ε are invariant under any local isometric transformation, and that the types I4ε,H1,H2 and H3ε are invariant if the first subsystem is transformed unitarily, while the second subsystem can be transformed by any isometry. More precisely, we prove below:

**Proposition** **1.**
*Any information quantity of type I1,I2 and I3ε as in Definition 4 is invariant under local isometric transformations of the form VAB(·)=VA⊗VB(·)VA†⊗VB†.*


**Proposition** **2.**
*Any information quantity of type I4ε,H1,H2 and H3ε as in Definitions 4 and 5 is invariant under local unitary transformations in the first system and isometric transformations in the second system of the form VAB(·)=UA⊗VB(·)UA†⊗VB†.*


The proofs for Propositions 1 and 2 use the technical Lemmas 1, 2 and 4 related to the isometric transformations and the corresponding so-called reversal channel [2], defined in the following. Due to VV†≠1 in general, the map V† is not necessarily trace-preserving, although it is completely positive. The reversal channel RV corresponding to an isometric channel V is a completely positive and trace-preserving map satisfying RV∘V=id, hence reversing the action of an isometric channel.

**Definition** **6**(Reversal channel)**.** *Let V:L+(H)→L+(H˜) be an isometric channel and ω∈D(H). The reversal channel corresponding to V and ω is defined as*(13)RV(σ):=V†(σ)+Tr[(1−VV†)σ]ω,
*for any σ∈L+(H˜).*

Note, that this channel is not unique and V(RV(ρ))≠ρ, in general. Also note that while σ∈L+(H˜), ω∈D(H) needs to be a quantum state such that the map is trace-preserving. Given an isometric channel V, we define the following sets that relate a smoothing environment to its image and pre-image under the isometric channel and its reversal channel:

**Definition** **7.**
*For a state ρ with ε-environment Bε(ρ) as in Definition 3, an isometric channel V, and the reversal channel RV, we define:*

(14)
BVε(ρ):={V(ρ^)|ρ^∈Bε(ρ)},


(15)
BRVε(ρ):={ρ˜|RV(ρ˜)∈Bε(ρ)}.



We now show that these sets can be ordered (see Figure 1), implying inequalities for smoothed information quantities as used in the proofs of Propositions 1 and 2.

**Lemma** **1.**
*Regarding the sets from Definitions 3 and 7, the following set relations hold:*

(16)
BVε(ρ)⊂Bε(V(ρ))⊂BRVε(ρ).



**Proof.** For the first relation let ρ˜∈BVε(ρ),ρ˜=V(ρ^),ρ^∈Bε(ρ). Using the invariance under isometric transformations (Equation 1), we have D(V(ρ)||ρ˜)=D(V(ρ)||V(ρ^))=D(ρ||ρ^)≤ε⇒ρ˜∈Bε(V(ρ)). Now let ρ˜∈Bε(V(ρ))⇒D(V(ρ)||ρ˜)≤ε. Using the data-processing inequality for any quantum channel, we have D(RV(V(ρ))||RV(ρ˜))≤ε⇒D(ρ||RV(ρ˜))≤ε⇒ρ˜∈BRVε(ρ). □

A second technical property of the reversal channels is used for proving invariance under local transformations. There exist reversal channels for local transformations that preserve the reversal property RV∘V=id locally, in the sense RV(VA(·)⊗(·))∼idA⊗(·).

For mutual information quantities of types I using quantum states as arguments of the divergence, the following fact is used for proving Proposition 1.

**Lemma** **2.**
*Let V=VA⊗VB be a local isometry. For any state ρA and any local quantum states σA,σB, there exists a reversal channel RV satisfying*

(17)
RV(VA(ρA)⊗σB)=ρA⊗TrA[RV(VA(σA)⊗σB)].



**Proof.** Define the reversal channel with ω=ρA⊗ωB for any local quantum state ωB. Using the definition of RV and VA†VA=1, one can directly calculate:(18)RV(VA(ρA)⊗σB)=ρA⊗(VB†σBVB+(1−TrB[VBVB†σB])ωB)=ρA⊗TrA[RV(VA(σA)⊗σB)].□

For conditional entropies of types H using both quantum states and positive-semidefinite operators as arguments of the divergence, the following Lemma is used for proving invariance in the proof of Proposition 2.

**Lemma** **3.**
*Let V=UA⊗VB with unitary UA and isometric VB. For any local quantum state σB, there exists a reversal channel RV satisfying*

(19)
RV(1A⊗σB)=1A⊗TrA[RV(πA⊗σB)],

*where πA=1dA1A is the maximally mixed state of A with dimension dA.*


**Proof.** Define the reversal channel with ω=πA⊗ωB for any local quantum state ωB. Considering UA†UA=UAUA†=1A, using the definition of RV and calculating as in the proof of Lemma 2, one finds:(20)RV(1A⊗σB)=1A⊗(VB†σBVB+(1−TrB[VBVB†σB])ωB)=1A⊗TrA[RV(πA⊗σB)].□

In the special case of smoothed states present in both arguments of the divergence as in I4ε, the following property is used for proving Proposition 2.

**Lemma** **4.**
*Let V=UA⊗VB with unitary UA and isometric VB. For any state ρA and any quantum state σAB, there exists a reversal channel RV satisfying*

(21)
RV(VA(ρA)⊗σB)=ρA⊗TrA[RV(σAB)].



**Proof.** Set ω=ρA⊗ωB for any quantum state ωB. Using the definition of RV and UA†UA=1, we then have for any local isometry V=UA⊗VB:(22)RV(VA(ρA)⊗σB)=ρA⊗VB†σBVB+Tr[1AB−VA(ρA)⊗VBVB†σB]ρA⊗ωB
(23)=ρA⊗(VB†σBVB+(1−TrB[VBVB†σB]ωB). On the other hand, using UAUA†=1A, one finds(24)ρA⊗TrA[RV(σAB)]=ρA⊗TrA[UA†⊗VB†σABUA⊗VB+Tr[(1AB−UAUA†⊗VBVB†)σAB]ρA⊗ωB]
(25)=ρA⊗TrA[1A⊗VB†σAB1A⊗VB+(1−Tr[1A⊗VBVB†σAB])ρA⊗ωB]
(26)=ρA⊗(VB†σBVB+(1−TrB[VBVB†σB])ωB),
showing the claimed equality. □

Using these results, we now prove Propositions 1 and 2.

**Proof of Proposition** **1.**Let V=VA⊗VB, where V(A/B):D(H(A/B))→D(H˜(A/B)). Let im(V(A/B))⊂D(H˜(A/B)) denote the image of V(A/B).First consider type I1. Using Tr(A/B)[VA⊗VB(ρAB)]=V(B/A)(ρB/A) and the invariance of any divergence under isometric evolution (Equation 1), one has:(27)I1(V(ρAB))=D(V(ρAB)||TrB[V(ρAB)]⊗TrA[V(ρAB)])
(28)=D(V(ρAB)||V(ρA⊗ρB))
(29)=D(ρAB||ρA⊗ρB)
(30)=I1(ρAB). Next, consider type I2. Again, using the invariance under isometric transformations, one has:(31)I2(ρAB)=infσB∈D(HB)D(ρAB||ρA⊗σB)
(32)=infσB∈D(HB)D(V(ρAB)||VA(ρA)⊗VB(σB))
(33)=infσ˜B∈im(VB)D(V(ρAB)||VA(ρA)⊗σ˜B)
(34)≥infσ˜B∈D(H˜B)D(V(ρAB)||VA(ρA)⊗σ˜B)
(35)=I2(V(ρAB)) Conversely, using the data-processing inequality for D with the reversal channel RV, one has:(36)I2(V(ρAB))=infσ˜B∈D(H˜B)D(V(ρAB)||TrB[V(ρAB)]⊗σ˜B)
(37)=infσ˜B∈D(H˜B)D(V(ρAB)||VA(ρA)⊗σ˜B)
(38)≥infσ˜B∈D(H˜B)D(ρAB||RV(VA(ρA)⊗σ˜B)) Using the reversal channel as in Lemma 2, with arbitrary σ^A∈D(HA), yields therefore:(39)I2(V(ρAB))≥infσ˜B∈D(H˜B)D(ρAB||ρA⊗TrA[RV(VA(σ^A)⊗σ˜B)])
(40)≥infσB∈D(HB)D(ρAB||ρA⊗σB)
(41)=I2(ρAB). Note, that the second inequality holds because TrA[RV(VA(σ^A)⊗σ˜B)]∈D(HB). The inequalities below (31) and (39) together imply the claimed equality.Finally, consider type I3ε. Using similar arguments and Lemma 1, one finds:(42)I3ε(ρAB)=infρ^AB∈Bε(ρAB)σB∈D(HB)D(ρ^AB||ρA⊗σB)
(43)=infρ^AB∈BVε(ρAB)σ˜B∈im(VB)D(ρ^AB||VA(ρA)⊗σ˜B)
(44)≥infρ˜AB∈Bε(V(ρAB))σ˜B∈D(H˜B)D(ρ˜AB||VA(ρA)⊗σ˜B)
(45)=I3ε(V(ρAB)) Conversely, using the reversal channel again as in Lemma 2 with arbitrary σ^A together with Lemma 1 and noting the set equality RV[BRVε(ρAB)]=Bε(ρAB), one concludes:(46)I3ε(V(ρAB))=infρ˜AB∈Bε(V(ρAB))σ˜B∈D(H˜B)D(ρ˜AB||VA(ρA)⊗σ˜B)
(47)≥infρ˜AB∈BRVε(V(ρAB))σ˜B∈D(H˜B)D(ρ˜AB||VA(ρA)⊗σ˜B)
(48)≥infρ˜AB∈BRVε(V(ρAB))σ˜B∈D(H˜B)D(RV(ρ˜AB)||RV(VA(ρA)⊗σ˜B))
(49)=infρ^AB∈Bε(ρAB)σ˜B∈D(H˜B)D(ρ^AB||ρA⊗TrA[RV(VA(σ^A)⊗σ˜B)])
(50)≥infρ^AB∈Bε(ρAB)σB∈D(HB)D(ρ^AB||ρA⊗σB)
(51)=I3ε(ρAB)□

**Proof of Proposition** **2.**Let V=VA⊗VB,VA(·)=UA(·)UA†, VA:D(HA)↦D(HA) with unitary UA and VB:D(HB)↦D(H˜B) with isometric VB. Let im(VB)⊂D(H˜B) denote the image of VB.First, consider the type I4ε. Using the invariance of D under isometric transformations, and Lemma 1, one has:(52)I4ε(ρAB)=infρ^AB∈Bε(ρAB)D(ρ^AB||ρA⊗ρ^B)
(53)=infρ^AB∈Bε(ρAB)D(V(ρ^AB)||V(ρA⊗ρ^B))
(54)=infρ˜AB∈BVε(ρAB)D(ρ˜AB||VA(ρA)⊗ρ˜B)
(55)≥infρ˜AB∈Bε(V(ρAB))D(ρ˜AB||VA(ρA)⊗ρ˜B)
(56)=I4ε(V(ρAB)). Conversely, using again Lemma 1 and applying the data-processing inequality for the reversal channel RV as in Lemma 4, one finds:(57)I4ε(V(ρAB))=infρ˜AB∈Bε(V(ρAB))D(ρ˜AB||VA(ρA)⊗ρ˜B)
(58)≥infρ˜AB∈BRVε(ρAB)D(ρ˜AB||VA(ρA)⊗ρ˜B)
(59)≥infρ˜AB∈BRVε(ρAB)D(RV(ρ˜AB)||RV(VA(ρA)⊗ρ˜B)). Now, since UA is unitary, Lemma 4 implies together with the set equality RV[BRVε(ρAB)]=Bε(ρAB),(60)I4ε(V(ρAB))≥infρ˜AB∈BRVε(ρAB)D(RV(ρ˜AB)||ρA⊗TrA[RV(ρ˜AB)])
(61)=infρ^AB∈Bε(ρAB)D(ρ^AB||ρA⊗ρ^B)
(62)=I4ε(ρAB),
proving the claimed property.Finally, consider the conditional entropy types H1,H2 and H3ε. Noting VA(1A)=1A due to the fact that VA is unitary, the proofs of local invariance are equivalent to those for I1,I2 and I3ε given in the proof of Proposition 1, respectively, if ρA is replaced by 1A and Lemma 3 is used instead of Lemma 2. □

## 4. Summary and Conclusions

In this work, we defined types of quantum information quantities based on generalized divergences and analyzed their properties regarding local isometric and unitary transformations. Leveraging properties of the reversal channel associated with the local transformation, we proved that several types of information quantities imply invariance under any local isometric transformation, while others are invariant if one of the subsystems is transformed by a unitary. Two main technical results are utilized to prove the local invariance. First, the smoothing environment required by some types of information quantities is related to its image and its pre-image under the isometric channel and a corresponding reversal channel, implying the set relation of Lemma 1. Second, the action of the reversal channel for a local isometric channel is characterized in Lemmas 2–4. Combining these insights with the data-processing inequality of any generalized divergence allows us to derive the main results regarding invariance for several types of information quantities under local isometric or unitary transformation as presented in Propositions 1 and 2.

Since methods for proving the invariance do not depend on the specific form of the generalized divergence used to define the information quantity, the invariances hold for numerous key quantities with operational relevance in quantum information processing. These invariances can enable more efficient computation of quantum information quantities by reducing complex states to simpler, equivalent forms with symmetries or smaller dimensions. For instance, consider the setting of quantum cryptography including a third party holding the purifying system of a bipartite state. If the output of a tripartite quantum channel is a pure state, it is equivalent to the purified output of a corresponding bipartite channel up to a local isometry in the purifying system. The dimension of the purification system of the latter state, however, may be smaller than the dimension of the former purification system. Consequently, the evaluation of relevant information quantities like the private information may be significantly improved by calculating it for the smaller but equivalent output state. Such invariances reduce computational overhead and unlock flexibility in protocol design. Given that the protocol performance is measured by an invariant information quantity, local unitary or isometric operations like encoding or correction operations can be added, modified or removed while preserving performance metrics.

Note, that the general proofs for local invariance depend only on the monotonicity by the data-processing inequality and the reversal properties of the isometric channel as given in Lemma 2–4. Therefore, local invariance may be shown similarly for other channels and corresponding reversals, such as the Petz recovery map (cf. [10]).

The computation of generalized information quantities remains a significant challenge, in general. While some quantities like the hypothesis testing mutual information can be calculated efficiently, e.g., by semidefinite programs [23,24], others can only be approximated (see Ref. [25] regarding the smooth max-relative entropy) or lack a general and efficient solution like the smooth max-mutual information. Our results may enable new methods to bound or approximate these quantities by leveraging state equivalences. Finally, we note that the invariance under local isometries, notably local Clifford and unitary operations, plays a pivotal role in the classification and optimization of graph states as used in error-correction [26,27] or measurement-based quantum computation [28,29], allowing for the optimization of codes or resource states with respect to an invariant information measure. In conclusion, the results of this contribution allow us to derive the behavior of general types of information quantities under local isometric or unitary transformations. Therefore, they improve the capability to characterize and compute information quantities relevant throughout the vast field of quantum information processing. 

## Figures and Tables

**Figure 1 entropy-27-01051-f001:**
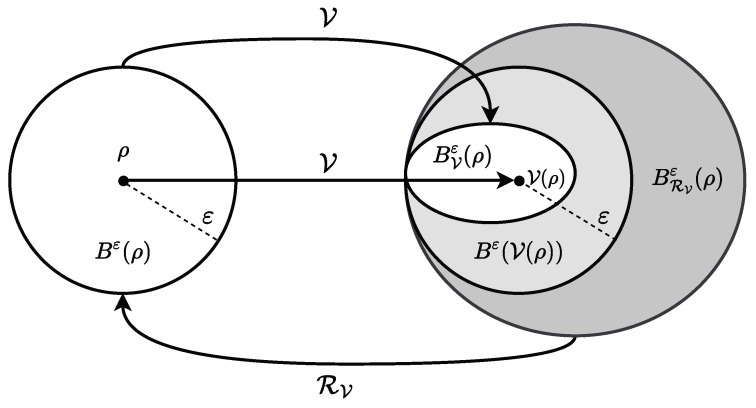
Schematic visualization of the smoothing environment Bε and its image BVε and pre-image BRVε under the isometric channel V and its reversal channel RV.

## Data Availability

No new data were created or analyzed in this study. Data sharing is not applicable to this article.

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
