# Peer review of "Local Invariance of Divergence-Based Quantum Information Measures"

_entropy, 2025, doi:10.3390/e27101051_

Round 1

Reviewer 1 Report

Comments and Suggestions for Authors

In this paper the authors prove local invariance properties of mutual and conditional information measures based on generalized divergences. They consider traditional, smoothed, and optimized variants, as well as combinations of these. They prove invariance under different forms of local isometries/unitaries.

The paper is very clear and well organized. The arguments are well written and therefore easy to follow, and as a result I have no doubt that the results are correct. The invariance of mutual/conditional information measures is an important topic in quantum information theory. The possibility of deriving these based on general divergences is both interesting and useful, and clarifies the essential structures underlying such properties. Therefore I find the paper interesting, and relevant to researchers in the field. The impact is sufficient for publication in Entropy, and the quality of the work is high.

For these reasons, I am happy to recommend the paper for publication in its present form. 

I do have some questions/suggestions, however these are matters of taste, and the authors are free to treat any changes as completely optional.

One question is about the choice to define the ϵ-balls based on the fidelity rather than (as in quant-ph/0512258 and 0803.2770) the trace distance. But in that light, isn't this paper a great opportunity to define the ϵ-balls in terms of an *arbitrary* generalized divergence? It seems this would fit the theme of the paper nicely, and the results would not change.

If the ϵ-balls are defined by any divergence, it seems to me the hierarchy in Fig. 1 automatically holds for any pair of CPTP maps. In particular, the image of the ball under any CPTP map is contained within the ball around the image (due to the contractivity), which is the main thing needed for the proofs.

Similarly, it seems to me the results could be strengthened while simultaneously simplifying the proofs.

In particular, I think that all four of the mutual information definitions are actually local-CPTP monotones.

The authors would have to double check the details, but it looks like you can show this fairly straightforwardly, just using the CPTP monotonicity, the property of images for divergence-based ϵ-balls, the properties of the infimum, and the commutation of partial trace with local channels.

If that's correct, the local CPTP monotonicity obviously implies that under any local CPTP V_A x V_B that is reversed by another local CPTP  R_A x R_B, you have invariance. 

I think that's already equivalent to being an isometry, so not much of an improvement, although you would at least get rid of needing the explicit form. But then you could also look at state-dependent reversals like the Petz recovery: for instance the first measure is invariant under any local CPTP that Petz recovers ρ from ρ given σ = ρA ⊗ ρB.

This seemed to look good for the mutual info measures, but apologies if I'm mistaken. I didn't look closely at the conditionals, but if the mutuals work then probably there is also something similar.

That said, if indeed this is incorrect, it might be helpful to point out in the text exactly which ones fail to be local CPTP monotones and why, as I probably wouldn't be the only one to wonder this.

Overall, congratulations on a nice and well written paper.

Reviewer 2 Report

Comments and Suggestions for Authors

The authors discuss a class of divergences  — measures of quantum statistical distinguishability — that obey invariance under stochastic (CPTP) maps.  This can be done without specifying the form of the underlying divergence measure. This is useful for communication and statistical tasks that need to satisfy such invariance but for which explicit demonstration is computationally challenging. 

The presentation is very technical and could be improved with a few extra sentences. For example,  simply stating that a reversal channel is CPTP map that approximately (or sometimes exactly) undoes the effect of another quantum channel,  would ease the non expert reader into the subject. The data processing inequality could be informally glossed as, no physically relevant operation can make two statistical distributions more distinguishable.  The value of the paper lies in the generality of its original proofs. It remains to be seen these  results enable new methods to bound or approximate the divergences.

The paper should be published as it is. 
